# Disulfiram—Mitigating Unintended Effects

**DOI:** 10.3390/antibiotics12020262

**Published:** 2023-01-28

**Authors:** Martha M. Grout, Kenneth B. Mitchell

**Affiliations:** 1Arizona Center for Advanced Medicine, Scottsdale, AZ 85258, USA; 2Nizhoni Functional Medicine, Scottsdale, AZ 85258, USA

**Keywords:** Lyme disease treatment, disulfiram toxicity, Borrelia burgdorferi treatment

## Abstract

Lyme disease caused by infection with a multitude of vector-borne organisms can sometimes be successfully treated in its very early stages. However, if diagnosis is delayed, this infection can become disseminated and, like another spirochetal infection syphilis, can affect multiple organ systems in the body, causing a wide variety of life-altering symptoms. Conventional antibiotic therapy may not be effective in eradicating the symptoms of the disease we know as Lyme disease. The recent literature has suggested that disulfiram (DSM) may be a potent drug in the armamentarium of physicians who treat chronic Lyme disease. The use of disulfiram in the treatment of Lyme disease started with a researcher who determined that DSM is bactericidal to spirochete. Encouraged by published case reports of apparent recovery from chronic Lyme disease, having prescribed DSM ourselves in the past for alcoholics who had a desire to stop drinking and prescribing it now for patients with chronic Lyme disease, we observed both predictable and potentially avoidable side effects not necessarily related to the ingestion of alcohol. We reviewed the published literature in PubMed and Google Scholar, using the following key words: Lyme Disease; Borrelia burgdorferi treatment; and disulfiram toxicity. This paper outlines the results of that research to help avoid some of the pitfalls inherent in this novel use of an old and established medication in the practice of clinical medicine.

## 1. Introduction

The recent literature has suggested that disulfiram (DSM) may serve as a potent addition to the armamentarium of physicians who treat chronic Lyme disease. Vector-borne diseases can, in theory, be treated in the early stages using a relatively short course of antibiotics, often doxycycline, amoxicillin or azithromycin, for a period of four to six weeks [1].

DSM is a thiuram derivative that blocks the oxidation of alcohol at the acetaldehyde stage, causing an elevation of aldehyde levels. Resulting symptoms due to excessive acetaldehyde may include flushing, nausea, vomiting, palpitations, chest pain, tachycardia, air hunger, weakness, hypotension, confusion and even death. Emergency physicians are warned to consider disulfiram overdose in alcoholic patients with unexplained symptoms [2].

When alcohol is consumed in the presence of DSM, serum acetaldehyde levels become elevated, causing a myriad of unpleasant and occasionally fatal symptoms [3]. In theory, in the absence of alcohol, these symptoms do not occur. Nevertheless, the literature on disulfiram describes both predictable and potentially avoidable adverse effects which may occur even in the absence of readily identifiable alcohol consumption. This research paper arises out of a desire to mitigate the adverse effects of DSM while still utilizing the benefits of the drug for Lyme disease patients.

## 2. History of Disulfiram

The use of DSM in treatment of Lyme disease started with the discovery that disulfiram is bactericidal to the Borrelia spirochete [4]. The drug had been used in the past to treat parasitic infections, such as tuberculosis and malaria [5,6,7], so its use for Lyme disease does not require a huge stretch of the imagination.

The first recorded case reports using DSM for the treatment of Lyme disease were published in 2019 [8].

Off-label use of commonly prescribed drugs is not unusual in the practice of medicine. Having used disulfiram in the past for alcoholics who had a desire to stop drinking made it seem reasonable to try something with prior history of success in our difficult-to-treat Lyme patients who were willing to experiment with a different approach.

It also seemed reasonable to conclude that if we could safely give the drug to alcoholics, there should be no problem giving it to patients with an infectious disease who were not alcoholics, in whom the possibility of adverse effects seemed remote. However, the onset of symptoms such as psychosis, uncontrollable babbling, uncontrollable dark thoughts, debilitating neuropathy and unexpected pain began to be reported in the Lyme chat groups. These symptoms had been reported in the earlier literature on disulfiram, associated with alcohol ingestion but not with the treatment of infectious diseases [3,9,10,11,12].

The history of DSM [13] starts with its synthesis by a German chemist M Grodzki in 1881, for reasons lost to the obscurity of history. A metabolite of DSM, carbon disulfide, was used in the rubber industry to hasten vulcanization—the cross-linking between rubber molecules through the sulfide bond improved the strength and consistence of the rubber. An article by an industrial physician, EE Williams, published in the JAMA in 1937 [12] noted the distressing effects of alcohol on workers in the rubber industry. In the 1940s, it was discovered that DSM was effective in the treatment of scabies and other parasites [13]. The drug was used to treat animals at first. Then, Dr. Jacobsen decided to take it himself, and very quickly realized that when he took the drug and drank any amount of alcohol, the effects were extraordinarily unpleasant. Thus, Antabuse^®^ was born. In the meantime, a large body of literature was published on the effect of disulfiram on multiple parasites, such as malaria and tuberculosis, as well as intestinal parasites.

In 2016, a group of researchers looking for new drugs to treat Lyme disease discovered the bactericidal effects of DSM [14]. One published case series used the drug for Lyme disease with good success [8]. Since then, the drug has been used in many patients who were previously not responsive to other commonly used antibiotics, such as doxycycline, azithromycin, metronidazole, amoxicillin/clavulanate, cephalosporins and others. There have been some successes, but the toxicity of DSM has become abundantly clear.

This toxicity may occur with or without the ingestion of alcohol, although certainly patients who do imbibe any form of alcohol find treatment with any dosage of disulfiram to be extremely unpleasant.

As such, if we are going to use DSM for its pathogen-killing effects, we need to be especially careful to inform our patients of the alcohol-containing (or even alcohol-like) supplements or drugs that they may already be taking. Homeopathic tinctures are often alcohol-preserved. Sugar alcohols (sorbitol, xylitol, erythritol, etc.) are still alcohols. Hand sanitizers are an especially ubiquitous source of hidden alcohol [15].

## 3. Mechanism of Action

The biochemical effects of DSM are excellently described in an article on Medscape, freely available for download [16]. DSM irreversibly inhibits the oxidation of acetaldehyde by competing with the cofactor nicotinamide adenine dinucleotide (NAD) for binding sites on aldehyde dehydrogenases [ALDH] [17]. The graphic for alcohol/NAD should be placed in this section. Uploaded already. If it needs a credit, that should be “Lucretia Alexandre”. Not published anywhere but in this article. An increased serum acetaldehyde concentration is thought to be responsible for the unpleasant symptoms of treatment with disulfiram, including:Headache,Tachycardia and palpitations,Hypertension,Nausea and vomiting,Anxiety and shortness of breath.

The directly toxic effects of DSM include neurologic [18], cutaneous [19] and hepatotoxic [20] sequelae. Indirect toxic effects may occur due to DSM’s inhibition of cytochrome P450 enzymes, resulting in a decreased clearance of drugs such as warfarin, phenytoin, benzodiazepines and some tricyclic antidepressants [21].

Approximately 20% of the drug remains in the body for 1–2 weeks post ingestion, significantly prolonging its effects due to the irreversible inhibition of ALDH and the body’s need to synthesize new stores of the enzyme.

DSM serves as a prodrug for diethyldithiocarbamate (DDC). DDC chelates copper, impairing the activity of dopamine beta-hydroxylase, an enzyme that catalyzes the metabolism of dopamine to norepinephrine, depleting presynaptic norepinephrine and causing an accumulation of dopamine. In the end, an accumulation of copper and oxidation of lipid membranes can result in myelin damage due to oxidative stress [22,23,24].

Nerve damage may be irreversible. In a rat study, peripheral nerve changes returned to control levels within a few weeks. However, the myelin changes in the brain were much more prolonged, perhaps because of continued accumulation of copper or changes in the levels of detoxifying proteins, such as superoxide dismutase 1 [25].

Neurotoxic effects associated with DSM include extrapyramidal symptoms. DSM administration to mice has resulted in lesions of the basal ganglia [26].

DDC inhibits superoxide dismutase (SOD), thereby impairing the ability to eliminate free radicals. DDC-induced methemoglobinemia can also occur secondary to the impairment (consumption) of glutathione-dependent methemoglobin reduction [23].

DSM can produce irreversible injury to the mitochondria by the oxidation of both glutathione and NAD [24], resulting in damage mainly to the organs with large numbers of mitochondria, i.e., brain, heart, muscle.

Given all the negative factors, does DSM have sufficient anti-spirochetal activity to warrant its use in Lyme patients? It appears that many microorganisms are susceptible to DSM. One hypothesis [27] attributes the immediate in vitro deceleration observed in bacterial growth to the cleavage of DSM by thiophilic components of intracellular metal ions, intracellular cofactors, such as coenzyme A reductase, and other enzymes within the microorganisms causing the depletion of glutathione and abrupt cessation of growth. This effect is seen in multiple microorganisms including giardia, mycobacteria, and staphylococci.

For many patients with chronic Lyme disease, the effects of the disease are worse than the more remote possibility of injury from a new and potentially helpful treatment.

As such, it behooves us, as their treating physicians, to ensure that such disulfiram treatment is as non-toxic as possible in the hopes that it will relieve patients of their intolerable symptoms of chronic Lyme disease.

## 4. Symptoms of Disulfiram Toxicity

It is important to distinguish disulfiram toxicity symptoms from symptoms of active neurologic infection and from Herxheimer reactions due to successful therapy.

Catatonia is reported in several articles [28,29,30]. Disulfiram-induced catatonia can be reversed simply by removing disulfiram from the system. Catatonia appears to be an idiosyncratic reaction because it does not occur in all patients and may recur when disulfiram is re-introduced.

Cytolytic hepatitis [31], ophthalmologic problems [32], cardiac abnormalities [33] peripheral neuropathy and encephalopathy [34,35,36,37] and even male sexual dysfunction and semen quality are reported to be toxic effects [32]. Hepatitis has been reported to progress to liver failure requiring transplantation [38,39].

Disulfiram is known to cause mitochondrial injury by inducing irreversible oxidation of both NAD (vitamin B3 derivative, nicotinamide adenine dinucleotide) and GSH (glutathione). Without these two crucial nutrients, mitochondria lose their transmembrane potential and become incapable of oxidative phosphorylation and energy generation [24].

Toxic effects may be delayed. One patient developed both neuropathy and encephalopathy after uneventful treatment with disulfiram for thirty years [34,35].

As always, drug–drug interactions may cause significant adverse effects [39].

Disulfiram metabolites may be at least partly to blame for some of the toxic effects. Cardiac abnormalities are described [33], as well as neuropsychiatric [37], ophthalmologic [32], sexual [40] and hormonal [41] adverse reactions.

Any symptoms arising in the liver, the heart or the nervous system (central or peripheral) after the institution of disulfiram therapy must be viewed through the lens of toxic suspicion, recognizing the difficulty of determining whether a symptom is due to Herxheimer reaction or DSM toxicity.

## 5. Common Drugs/Pharmaceuticals/Supplements to Avoid

Some common over-the-counter medications that contain an ethanol concentration greater than 5% include Adult Tylenol^®^ liquid, Benadryl^®^ Elixir, Comtrex, Donnatal^®^ Elixir, Dramamine^®^ Liquid, Geritol^®^ Liquid, NyQuil^®^ Liquid, Formula 44 Cough Mixture^®^ and Tylenol and Codeine Elixir^®^. Mouthwash and facial cleaning products may also contain alcohol. Hand sanitizers are a notorious source of hidden alcohol.

Others contain sugar alcohols added to “no added sugar” products, such as ice cream and sauces, as well as things such as sorbitol, xylitol, erythritol and foods promoted for diabetics.

Many other agents may, in the presence of alcohol, produce a DSM-like reaction, such as the following:Industrial solvents such as trichloroethylene (the degreaser’s flush) [42],Mushrooms (e.g., *Coprinus atramentarius* (inky cap), *Clitocybe claviceps*),Antibiotics (e.g., metronidazole, sulfonamides, some cephalosporins, nitrofurantoin, chloramphenicol),Pesticides (e.g., carbamates, monosulfiram (Tetmosol)),Chloral hydrate,Antifungals (griseofulvin).

## 6. Mitigation of Disulfiram Toxicity

In addition to the obvious course of stopping disulfiram therapy, there are multiple avenues available to mitigate some of the adverse or “side” effects of disulfiram.

The “irreversible mitochondrial injury” caused by the irreversible oxidation of glutathione and NAD can be at least mitigated and potentially reversed by giving enough glutathione and NAD to replenish the mitochondrial stores.

### 6.1. Glutathione

All mammalian systems use glutathione for detoxification. Disulfiram induces the irreversible oxidation of both glutathione and nicotine adenine dinucleotide (NAD).

One of the most concerning metabolic effects of disulfiram is its depletion of glutathione through an irreversible bond, depleting the body of glutathione and potentially resulting in irreversible toxicity. The mechanism lies in disulfiram’s ability to oxidize oxyhemoglobin to methemoglobin, resulting in (a) tissue hypoxia and (b) generation of hydrogen peroxide, a cellular oxidizing compound [43]. Peroxide is then metabolized to water through the consumption of glutathione. If there is a pre-existing depletion of glutathione, this could explain the differential toxic effect of DSM in different individuals. If the glutathione levels can be restored, the disulfiram administration should result in less toxicity.

DSM impairs the permeability of the mitochondrial inner membrane. Mitochondria use both NAD and glutathione to metabolize disulfiram. If the mitochondrial membrane is damaged, the mitochondrial membrane loses its electrochemical gradient, resulting in the inhibition of oxidative phosphorylation and insufficient production of energy in the form of ATP [24]. The end result is the loss of the mitochondrial transition pore, allowing the free passage of molecules in and out of the mitochondria, resulting in ATP depletion and cellular apoptosis [44].

### 6.2. Sarsaparilla—Smilax

This plant grows all over the world and is used as an anti-inflammatory agent for diseases such rheumatoid arthritis [45], respiratory syncytial virus, Herpes simplex virus [46] and skin conditions such as psoriasis [47]. It has even been shown to have anti-fungal activity against Candida species [48]. Its therapeutic use in secretory diarrhea and dysentery has been described [49]. Smilax is given orally in capsule or powder form, and it is generally available through suppliers of oriental herbs [46,49,50].

The mechanism of action is thought to be through its antioxidant actions [51].

As an interesting aside, Smilax can also chelate iron, using a chemical similar to something in apples—the one that inhibits intestinal uptake of glucose through inhibition of the sodium-dependent glucose transporter [52]. It even protects against heavy-metal-induced toxicity [53].

### 6.3. Dihydromyricetin

Dihydromyricetin (DHM) is a compound found in traditional Chinese “vine tea” preparation, long used in oriental medicine for its calming anxiolytic properties, as well as for its ability to mitigate the effects of a hangover [54], Patients with Lyme disease often have neurologic and psychiatric symptoms which can mimic a hangover, such as brain fog, anxiety, headache and peripheral neuropathy, which are conventionally treated with anxiolytics, pain medications and GABA agonists.

DHM potentiates GABA receptors, reducing the effect of alcohol on those receptors. Flumazenil, a benzodiazepine antagonist, blocks the effect [54].

Alcohol-induced liver toxicity results in increased proinflammatory cytokines, oxidative stress, and the activation of Kupffer cells. In a mouse study, alcohol ingestion resulted in lipid accumulation, reducing fatty liver and liver triglycerides. DHM reduced both alcohol and acetaldehyde concentrations, as well as the expression of proinflammatory cytokines [55].

How does the dihydromyricetin effect on alcohol dependency relate to the unpleasant effects of disulfiram? Disulfiram treatment can also result in some of the same unpleasant effects.

One of the major side effects of disulfiram is neurologic—either neuropathy or severe anxiety. DHM treats anxiety extremely well through its effect on the GABA receptors [56]. If it works so well in mice and has been used in Chinese medicine for several thousand years, it seems reasonable to give it to our patients with those symptoms. Dihydromyricetin would be a natural choice for the mitigation of the inflammatory symptoms induced by treatment of Borrelia burgdorferi infection with DSM. Phase 1 studies have already demonstrated the safety of DHM through its use in traditional Chinese medicine.

Oxidative stress is a major cause of neurotoxicity. DHM has been found to stimulate levels of NADP (nicotinamide adenine dinucleotide phosphate). DHM also has significant antioxidant activity through the activation of the Nrf2 pathway [57]. In addition, the compound has been shown to reverse neuropathologic changes in the brains of demented mice by “reducing Aβ peptides while restoring gephyrin levels, GABAergic transmission and functional synapses” [58]. Admittedly, mice are not human beings, but they are certainly on the continuum.

It is recommended to NOT combine DHM with other flavonoids, as other flavonoids have been shown to reduce the efficacy of DHM [59].

### 6.4. NAD—Nicotinamide Adenine Dinucleotide

DNA, the genetic information contained in every one of our cells, is constructed from nucleotides. RNA, the messenger that transmits the instructions encoded in our DNA to our peripheral cells, is constructed from slightly different nucleotides. The following three other nucleotide compounds are present in our bodies: ADP (adenosine diphosphate, the precursor to the ATP molecule that runs our cellular engine); ATP (adenosine triphosphate, the required molecule of energy); and NAD (nicotinamide adenosine dinucleotide), the molecule that facilitates communication between RNA and the cell’s proteins.

Nucleotides are the subunits which form the nucleic acids ribonucleic acid (RNA) and deoxyribonucleic acid (DNA), serving as the cell’s storehouse of genetic information. Nucleotides are essential for the proper growth of rapidly proliferating cells.

Niacin, vitamin B3, is an essential vitamin. Deficiency of niacin in its most extreme form results in a disease called pellagra, characterized by the four Ds—dermatitis, diarrhea, dementia, and eventual death. Food sources include animal proteins, fish, “fortified” cereals and breads, seeds and nuts, and beans and bananas [60,61].

Niacin (vitamin B3) occurs in the body as two metabolically active coenzymes, NAD (nicotinamide adenine dinucleotide) and NADP (NAD phosphate).

The niacin cofactor NAD and coenzyme NADP have major roles in energy-related and biosynthetic metabolic processes [60].

Independent of its functions as a coenzyme, vitamin B3 or niacin is also involved in the regulation of normal blood lipoprotein and cholesterol levels, and the maintenance of normal vascular tone.

NAD is required in over five hundred enzymatic reactions in the body, playing a role in a multitude of the body’s biological processes [61]. Nerve cells can import active NAD+, while all other body cells must synthesize it from scratch using tryptophan and nicotinic acid or nicotinamide as a substrate. Is it any wonder that smokers insist that their use of tobacco enhances their brain function? Niacin derived from food intake recycles NAD+ within the cell [62].

The two main human NAD+ responsive signaling protein families are the sirtuins and the PARPs. Sirtuins are telomere-protective proteins that regulate mitochondrial metabolism, inflammation, circadian rhythms, and cell death. The PARPs (poly-ADP ribose polymerase) are enzymes essential for DNA repair and transcription of RNA from DNA. When the PARPs are activated, cells become depleted of NAD+ and eventually die, unless the NAD is replaced.

Humans are not the only organisms to use NAD. NAD is an essential element for all living organisms, from one-celled organisms such as yeasts to multicellular organisms such as mammals [63].

The production of NADH does not appear to be affected by age—however, older cells consume more NADH and glutathione, leading to a loss in detoxification reserve—which one researcher calls a catastrophic decline in redox ratio [64]. Older neurons (at least in rats) appear to synthesize as much NADH as younger neurons but consume more of the same substance when under stress [64], resulting in less resilience and less reserve against further stress. There is evidence to suggest that supplementation may improve the cell redox status, thus protecting against further damage and improving the response to stress [65].

Dietary niacin is well absorbed and taken up by the liver, with any excess being metabolized by the liver and excreted by the kidneys. Food sources include meats, fish, avocados, peas, brown rice and sunflower seeds, for example.

Patients with chronic fatigue syndrome may significantly benefit from NAD+ therapy. An excellent review published in 2020 discusses NAD+ metabolism and its therapeutic potential [66]. Clinical experience has shown even better results if accompanied by glutathione.

To better understand the function of NAD+ in mitochondrial metabolism, it is important to understand that NAD serves both as a stimulant for the production of energy and as a regulator of that very energy production [67]. If the organism is in a state of extreme fatigue, such as a marathon runner who collapses at the finish line, with a near total depletion of both ADP and ATP, NAD+ could well be used as an emergency source for ATP synthesis [68].

Most recently, a review of the clinical evidence for the therapeutic use of NAD was published in the journal *Pharmaceuticals* [69]. Studies have been conducted in the treatment of psoriasis and the enhancement of skeletal muscle activity. Published case reports discuss the use of NAD in patients with Parkinson’s disease [70].

Boosting NAD+ levels can result in lifespan increase in yeast cells as much as exposure to minor stresses, such as heat and electrolyte concentration. The overexpression of an enzyme that increases NAD+ levels in fruit flies results in a 30% increase in lifespan. For humans, that increase would result in another 22 years of active life.

Nicotinic acid has been known since the mid-1950s to reduce cholesterol levels [71]. Unpleasant skin sensations—burning and itching—have limited its use. Most people better tolerate nicotinamide.

One recent article published in Medical Hypotheses discusses the role of NAD (and zinc) in controlling cytokine storm, as well as endothelial activation and coagulability.

Human studies have been performed using NA (nicotinamide riboside). Numerous safety and efficacy studies address oral supplementation with nicotinamide riboside [65,69,71,72,73,74,75,76,77].

One safety study using products found in milk (NA) and blueberries (pterostilbene) demonstrated sustained NAD+ levels for at least the 8-week duration of the study [76].

An excellent review of the therapeutic potential for NAD was published in 2021 [71].

How can we improve NAD levels in our patients? NAD may be taken orally using commercially available supplements. 

-Nicotinamide Riboside (Tru Niagen^®^) is a proven well absorbed and well tolerated form that raises NAD+ [72,77]-Niacin and an extended-release version (Niaspan^®^) have been tested in human subjects and have been shown to raise NAD+ levels with oral doses of 1000 mg per day, resulting in one trial in a two-fold increase in NAD+ levels after 7 days. Clinical trials are under way using NMN (nicotinamide mononucleotide), currently available for purchase on the Internet. Very little else is available beyond articles speculating on potential usefulness.

NAD may also be administered intravenously. One published study used a dose of 750 mg given intravenously over 7 h. That long of an infusion is a little impractical for a clinical office, although it would certainly be easy enough to do in hospital using an infusion pump.

Other forms of NAD are available through compounding pharmacies. Some of these may be administered a little more rapidly; although, in our experience, slower at the start ends up being faster eventually. A different method for the preparation of NAD infusion products uses cyclodextrins as a vehicle to encourage more rapid incorporation into the cell [78].

### 6.5. R-Lipoic Acid

Lipoic acid in its reduced form (dihydrolipoic acid) is catalyzed by NAD-dependent enzymes. In rats, lipoic acid has been shown to reduce cardiac arrhythmias and cardiac-related mortality through the restoration of aldehyde dehydrogenase activity [79]. Lipoic acid is a physiologic dithiol that acts in a similar manner as dithiothreitol, commonly used in chemical experiments to prevent the inhibition of aldehyde dehydrogenase. Lipoic acid (also known as alpha-lipoic acid) exists as a racemic enantiomer; the S-enantiomer is the stronger inhibitor, and the R-enantiomer is the better activator of dehydrogenase enzymes [80]. Thus, the R-enantiomer is preferred over racemic ALA/lipoic acid as it will not have the enzyme-inhibiting S-form.

Reports of lipoic acid’s usefulness in the treatment of oxidative stress appear in the literature as early as 1998 [81]. Lipoic acid’s major effect appears to be through its redox activity, serving as both an oxidizing and anti-oxidizing molecule, depending on the needs of the cell. Disulfiram creates enormous oxidative stress on cells, in the presence of alcohol. R-lipoic acid, given either orally or intravenously, helps to mitigate the stress [82].

### 6.6. Melatonin

Melatonin is an ancient antioxidant [83]. Acutely administered melatonin has been shown to rescue oxidatively stressed mitochondria. [83,84,85,86,87,88,89,90].

### 6.7. Desmodium

The herb Desmodium (molliculum or adscendens) aka Burbur or Amor Seco has been used traditionally for its body-cleansing properties. It is very effective in aiding the detoxification of the liver, kidneys and lymphatic system [91,92,93,94,95].

Active plant ingredients, including vitexin and D-pinitol, have been found to have strong antioxidant activity in vivo [94] and are reported to be strongly protective of the liver. This plant has been used in folk medicine for centuries. In vitro studies showed that Desmodium inhibited lipid peroxidation and scavenged hydroxyl and superoxide radicals [96]. The antioxidant and free radical scavenging properties of Desmodium are the most significant in the mitigation of DSM-induced symptoms.

It will be important, obviously, to use a water extract or capsule form, not an alcohol extract of the herb.

## 7. Conclusions

Disulfiram is a new weapon in our therapeutic armamentarium used in the treatment of these devastating vector-bone infections, known collectively and colloquially as Lyme disease. Disulfiram is an old pharmaceutical medication recently re-purposed as a bactericidal antibiotic to treat patients with symptoms of chronic Lyme disease unresponsive to the IDSA-recommended regimen of a two-week course of doxycycline or to the ILADS-recommended regimen of a more prolonged therapy with antibiotics.

Disulfiram treatment results in unintended side effects in some people, including symptoms such as agitation, dysphoria, peripheral tingling and numbness. Mitigation strategies are outlined, based on review of the literature and in consideration of the mechanisms of action of disulfiram.

In general, substances which reduce toxins and cellular or mitochondrial damage result in a reduction in the release of histamines, cytokines and other pro-inflammatory molecules, and therefore a reduction in the symptoms caused by disulfiram-induced mitochondrial stress. Substances which restore function to damaged mitochondria include herbs such as sarsaparilla (smilax), dihydromyricetin and desmodium. Antioxidant co-factors which can also restore mitochondrial function include glutathione, NAD plus and lipoic acid.

The preventive use of substances described in this article can help to make the treatment of Lyme disease with disulfiram both more rapidly effective and less fraught with adverse effects.

## Data Availability

All the references are to be found in the journals specified, or online at the links specified.

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
