# Peer review of "Disulfiram—Mitigating Unintended Effects"

_antibiotics, 2023, doi:10.3390/antibiotics12020262_

Round 1

Reviewer 1 Report

The Grout and Mitchell paper submitted to Antibiotics seems very interesting but incomplete for publication in its current form. The authors describe the toxic effects of disulfiram and how the effects can be mitigated or even eliminated. However, in the summary of the work, the need to identify and mitigate the adverse effects of disulfiram due to its repositioning to treat Lyme disease is raised.

This raises the need to describe the mechanism of action or proposed mechanism by which disulfiram eliminates Borrelia burgdorferi. Taking into account the processes involved in eliminating the bacteria, it is possible to establish, based on the doses used. These main adverse symptoms may occur and focus the discussion on those conditions in particular.

The approach presented in the manuscript currently only covers the general toxicology of disulfiram but does not directly involve the conditions used to treat Lyme disease, which is the central message of the abstract.

I recommend that authors emphasize the proposed mechanisms of action to kill the bacterium responsible for Lyme disease using disulfiram to have the desired impact and to differentiate the work from many other recently published papers describing the adverse effects and toxicology of disulfiram.

Author Response

We will review what has been documented for mechanisms of action of DSM on other infections, if any. 

We have incorcorporated the requested information on possible mechanism of action of DSM or microbial mitochondria and cellular metabolic processes. 

Reviewer 2 Report

The review article “Disulfiram – mitigating unintended effect discussed the Disulfiram [DSM] is a potent drug in the armamen- 7 tarium of physicians who treat chronic Lyme disease. It is well- a written, emerging topic, but I feel that the article needs a few modifications before acceptance.

1.      Use some English editing software, there are some minor grammatical mistakes are in manuscript.

2.       In Paragraph 3, use some diagrammatical representation for understand mechanism of action of Disulfiram.

3.       In Paragraph 5 , discussed in detail about drugs used for Lyme treatment in comparison with disulfiram.

4.       Write in detail about the How does the dihydromyricetin effect on alcohol dependency relate to the unpleasant effects of disulfiram?

5.       How can we boost NAD levels in our patients? Use some diagrammatic representation.

6.       Finally in the last paragraph author may also discuss the challenges and opportunities for different effects in detail. The paper should be minor revision is in order before this work can be further considered for publication.

Author Response

Thank you,  good suggestions, we will work on them. 

We did make a diagrammatic representation of mechanism of action of disufiram as it relates to alcohol metabolism. 

I do not always agree with the microsoft editor. I made changes where appropriate, but left some because to change would make things less clear, or more verbose with more convoluted sentences. 

Thank you for your time and effort in reviewing the paper. 

Reviewer 3 Report

The manuscript summarizes the uses and side effects of the drug known as Disulfiram. The manuscript has nicely written and it can be accepted after major revision.

1.      Authors should revise the complete manuscript in terms of grammar as well as the English language wise (major revision).

2.      Abstract needs to be improved scientifically.

3.      First and second line of the abstract and introduction is the same. It should be changed.

4.      Line number 124-125 “Neurotoxic effects associated with DSM include extrapyramidal symptoms, and lesions of the basal ganglia have been described in patients after therapy with DSM” should be changed or modified.

5.      Line no. 149-150 should be changed.

6.      Please elaborate line no 157-158.

7.      Please elaborate the role of Smilax in terms of mitigating DSM toxicity.

8.      Elaborate on the role of desmodium.

9.       Conclusion content should be increased.

Author Response

We have done our best to address all the issues raised in the comments. 

Round 2

Reviewer 1 Report

Probably, I needed to explain myself correctly last time.

In the paper's abstract, the authors describe different strategies used to reduce the already known toxic effects of disulfiram (DSM) to show the possible advantages of its use in the treatment of Lyme disease, according to recent reports in the literature.

The information contained in the current document, the toxicity of disulfiram and the different strategies to reduce it, has already been reported on other occasions. What might make the paper submitted by Grout and Mitchell eligible for publication in Antibiotics would, in my view, be the description of the advantages and disadvantages of the particular use of disulfiram in the treatment of Lyme disease.

For example, the description of the antibacterial activity of disulfiram against the spirochete responsible for Lyme disease in comparison with FDA-approved drugs (Drug Des Devel Ther. 2016 Apr 1;10:1307-22. doi: 10.2147/DDDT .S101486. eCollection 2016. Identification of new drug candidates against Borrelia burgdorferi using high-throughput screening), In vitro and in vivo evaluations of disulfiram identifying advantages and disadvantages (Antibiotics 2020, 9(9), 633; https://doi.org/10.3390/antibiotics9090633. Repurposing Disulfiram (Tetraethylthiuram Disulfide) as a Potential Drug Candidate against Borrelia burgdorferi In Vitro and In Vivo), and other studies that clearly identify the potentiality of disulfiram as a therapeutic alternative to Lyme disease.

In its current form, just one sentence (lines 217-219) deals with the above description: Dihydromiracetin would be a natural 217 choice for mitigation of the inflammatory symptoms induced by treatment of Borrelia 218 burgdorferi infection with DSM. 

The manuscript, in its current state, does not constitute a contribution to the study of disulfiram toxicity or strategies to mitigate it. But above all, it does not contribute in any way to identifying disulfiram as a therapeutic strategy to combat Lyme disease, as the authors claim in the summary and conclusions of their work.

Author Response

It appears that reviewer #1 actually wishes us to write a different article – giving the pros and cons for the use of disulfiram in the treatment of Lyme disease.

The article is intended to help those clinicians who have already made the decision to use DSM in a patient – because other therapies have been unsuccessful or caused too many unwanted effects. DSM is an old drug, but it has not until recently been used for anything other than to make an alcoholic think twice about picking up the first drink of the day. Granted that it was used in years gone by for treatment of infectious diseases. That was in the years before I went to medical school in the 1970s, so DSM has not been used as an antibiotic in modern medicine.

When we, as clinicians, learned about the potential use of DSM in our incompletely treated Lyme patients, we were excited to have another weapon in our tool kit. We rapidly learned about the disadvantages of that weapon, and did research on ways of mitigating the disadvantages, so that we might continue to use the therapy, the DSM.

We thought that we might not be alone in the realization of the difficult and unpleasant side effects, and wrote this article in an attempt to share our research.

The article does not purport to:

  • Study disulfiram toxicity – only to consolidate and describe what has previously been reported
  • Identify disulfiram as “a therapeutic strategy to combat Lyme disease” – that has already been described and the articles are included in the references – which include those articles that reviewer #1 mentions in the critique.
  • describe the advantages and disadvantages of the particular use of disulfiram in the treatment of Lyme disease – again, that has already been done in the above-mentioned references.

The article DOES aim to give some strategies to mitigate untoward side effects of DSM, after a clinician has already chosen to use the drug in a Lyme disease patient. The article was written because we did not find any such information collected in one place, or one article, in the current literature. So we do take exception to the comment that the article does not contribute in any way to strategies designed to mitigate the potential toxicity of disulfiram.

Reviewer 3 Report

The manuscript has been revised significantly and can be accepted.

Author Response

Thank you, we will triple check the MS.